# Early Myocardial Changes in Patients with Rheumatoid Arthritis without Known Cardiovascular Diseases—A Comprehensive Cardiac Magnetic Resonance Study

**DOI:** 10.3390/diagnostics11122290

**Published:** 2021-12-07

**Authors:** Ewa Malczuk, Witold Tłustochowicz, Elżbieta Kramarz, Bartłomiej Kisiel, Magdalena Marczak, Małgorzata Tłustochowicz, Łukasz A. Małek

**Affiliations:** 1Department of Internal Diseases and Rheumatology, Military Institute of Medicine, 04-141 Warsaw, Poland; wtlustochowicz@wim.mil.pl (W.T.); bkisiel@wim.mil.pl (B.K.); mtlustochowicz@wim.mil.pl (M.T.); 2Department of Cardiology and Internal Diseases, Military Institute of Medicine, 04-141 Warsaw, Poland; ekramarz@wim.mil.pl; 3MR Unit, Department of Radiology, National Institute of Cardiology, 04-635 Warsaw, Poland; mmarczak@ikard.pl; 4Department of Epidemiology, Cardiovascular Disease Prevention and Health Promotion, National Institute of Cardiology, 04-635 Warsaw, Poland; lmalek@ikard.pl

**Keywords:** rheumatoid arthritis, cardiovascular magnetic resonance, parametric imaging, mapping, myocardial oedema

## Abstract

Clinically silent cardiac disease is frequently observed in rheumatoid arthritis (RA), and cardiovascular complications are the leading cause of mortality in RA. We sought to evaluate the myocardium of young RA patients without known cardiac disease using cardiac magnetic resonance (CMR), including T1/T2 mapping sequences. Eighteen RA patients (median age 41 years, 83% females) mainly with low disease activity or in remission and without any known cardiovascular disease were prospectively included to undergo CMR. A control group consisted of 10 sex- and age-matched patients without RA or any known structural cardiovascular disease. Heart chambers size and left/right ventricular systolic function were similar in patients with RA and controls. Signs of myocardial oedema were present in up to 39% of RA patients, including T2 time above cut-off value in 7 patients (39%) in comparison to none of the controls (*p* = 0.003) and T2 signal intensity ratio above the cut-off value in 6 patients (33%) and in none of the controls (*p* = 0.06). Extracellular volume was similar in both groups signifying a lack of diffuse fibrosis in studied group of RA patients. There were also no signs of late gadolinium enhancement (LGE) in either group except for one patient with RA who was found to have prior silent myocardial infarction. No correlation was found between markers of disease severity and markers of oedema observed on CMR in patients with RA. Nevertheless, patients with increased T2 time (≥50 ms) were more likely to have X-ray erosions (*p* = 0.02) and a longer duration between symptom onset and diagnosis (*p* = 0.02). Finally, there were no significant arrhythmias on 24-h ECG Holter monitoring in RA patients. CMR features of myocardial oedema without signs of myocardial fibrosis were found in 39% of young RA patients without known heart disease or cardiac symptoms. Presence of myocardial oedema was associated with X-ray erosions and a longer duration between symptom onset and diagnosis. The clinical significance of the observed early myocardial changes accompanying RA requires additional studies.

## 1. Introduction

Rheumatoid arthritis (RA) is a systemic connective tissue disease that affects approximately 1% of the population [1]. The onset of the disease usually occurs between 30–50 years. The hallmark of RA is arthritis, but the disease can also affect other organs and systems, such as the respiratory system (pleurisy, pulmonary fibrosis), kidneys, or cardiovascular system (pericarditis, endocarditis, myocarditis). Rheumatoid arthritis, like many other chronic inflammatory diseases, causes accelerated development of atherosclerosis. Cardiovascular complications are the leading cause of mortality in RA patients, accounting for over 50% of deaths [2,3,4]. For this reason, studies assessing the prevalence and nature of morphological and functional changes in the cardiovascular system in RA are of significant clinical importance. Clinically silent cardiac disease is frequently observed in RA patients [5]. Echocardiography is the primary tool in the assessment of cardiac morphology and function. However, cardiovascular magnetic resonance (CMR) has been shown to detect changes at an earlier stage, and many authors point out the potential utility of CMR in the diagnosis of heart disease in RA, especially in the diagnosis of early, subclinical disease [6,7,8,9]. Early recognition enables the prophylaxis and early treatment of cardiac disease and, consequently, can significantly improve the prognosis in this group of patients [6]. Previous studies have presented subclinical morphological changes of the heart (fibrosis, inflammation) using CMR in patients with RA [6,7,8,9].

The latest asset in visualization of myocardial changes in CMR is the implementation of new resonance sequences for T1 and T2 time mapping, which enables non-invasive analysis of tissue changes in the myocardium [10]. These new sequences, also referred to as parametric imaging, allow the assessment of myocardial oedema (T1 and T2 mapping) and diffused fibrosis (T1 mapping before and after contrast agent administration). T1 time evaluation pre- and post-contrast administration allows calculation of extracellular volume (ECV) of the myocardial tissue increased in generalised fibrosis. This technique has been strongly correlated with histological findings [11]. So far, studies with the use of parametric imaging in patients with RA are very limited and restricted almost completely to T1mapping without the use of T2mapping and its advantages in detection of myocardial oedema [8,12]. Additionally, CMR studies were not performed on patients at early stages of RA, which could demonstrate the broader spectrum of potential myocardial changes during the full course of the disease.

The aim of our study was to evaluate the myocardium of young RA patients without known cardiac disease using CMR, including T1/T2 mapping. In addition, we aimed to assess the correlation between myocardial changes on CMR, the baseline characteristics of the study group, and functional changes as assessed by electrocardiogram (ECG) Holter monitoring.

## 2. Materials and Methods

### 2.1. Study Group

Eighteen RA patients without known cardiac disease were prospectively recruited from the Internal Medicine and Rheumatology Department and outpatient clinic at the Military Institute of Medicine in Warsaw between November 2019 and August 2020. Inclusion criteria consisted of age greater than 18 years but less than 50 years (to limit the influence of age-dependent myocardial changes) and the ability to provide informed consent and meeting of American College of Rheumatology (ACR)/European League Against Rheumatism (EULAR) criteria for RA (modified in 2010) [13]. Exclusion criteria included known cardiovascular disease, hypertension, symptoms of cardiovascular diseases, diabetes mellitus, chronic kidney, lung or liver diseases, cancer, and contraindications to CMR. Based on medical history, basic information on the underlying disease and comorbidities was collected. Laboratory tests were measured in the hospital laboratory. The normal level of C-reactive protein (CRP) was <0.8 mg/dL, and the erythrocyte sedimentation rate (ESR) was <12 mm/h. Rheumatoid factor (RF) and anti-cyclic citrullinated peptide antibody (ACPA) were defined as seropositivity above the upper limit of normal. High ACPA/RF titres were defined as exceeding three times the upper limit of normal [13]. We assessed RA patients for disease severity to see if there was a correlation between disease activity and myocardial changes on CMR. Disease activity in RA was measured using the Disease Activity Score with 28 joints (DAS28), which includes the number of swollen and tender joints, patient’s global health score on a visual analogue scale (VAS), and CRP (DAS28-CRP) or ESR(DAS28-ESR) [14,15]. The level of RA disease activity was interpreted as low (DAS28 ≤ 3.2), moderate (3.2 < DAS28 ≤ 5.1), or high (DAS28 > 5.1). A DAS28 < 2.6 corresponded with being in remission. Joint destruction was evaluated by radiograms of both hands and feet as the presence of bone erosions and by Steinbrocker’s grading, assessed by a single, trained radiologist blinded to other patients’ data [16]. Functional status in RA was measured by Health Assessment Questionnaire Disability Index (HAQ) [17]. 

### 2.2. Cardiac Magnetic Resonance

The study was performed with the use of a Siemens MagnetomAvanto Fit 1.5 Tesla scanner (Siemens, Erlangen, Germany). The protocol included initial scout images, followed by cine balanced steady-state free precession (bSSFP) breath-hold sequences in 2-, 3-, and 4-chamber views. Short axis was identified using the 2- and 4-chamber images and a stack of images was acquired, which included the ventricles from the mitral and tricuspid valvular plane to the apex.

Pre-contrast T1mapping with modified Look Locker sequence (MOLLI) and T2mapping with a T2-prepared SSFP sequence were performed immediately after acquisition of the bSSFP cine images and processed using MyoMaps software (Siemens, Erlangen, Germany). For that purpose, 3 short axis slices (one basal, one mid-ventricular, and one apical) and 2-, 3-, and 4-chamber views were obtained. Subsequently, an acquisition of dark-blood T2-weighted (T2W) images with fat suppression in the same orientations was performed. Following these acquisitions, 0.1 mmol/kg of a gadolinium contrast agent (gadobutrol—Gadovist^®^, Bayer SheringPharma AG, Berlin, Germany) was administered and flushed with 15 mL of isotonic saline. Late gadolinium enhancement (LGE) images in 3 long axis and a stack of short axis imaging planes were obtained with a breath-hold phase-sensitive inversion recovery sequence (PSIR) 5–15 min after the contrast injection. The inversion time was adjusted to null normal myocardium (typically between 250 and 400 ms as assessed by means of a TI-scout acquisition). This was followed by a post-contrast T1mapping acquisition 15 min after the contrast injection in the same orientations as the pre-contrast T1mapping.

Images were analysed by a cardiologist and radiologist, both with 13 years of experience in CMR blinded to other patient data (Ł.A.M., M.M.) and with the use of dedicated software (Syngovia, Siemens, Erlangen, Germany). End-diastolic and end-systolic endocardial and epicardial contours were drawn manually for the left and right ventricle in the short axis stack of bSSFP cine acquisitions. Delineated contours were used for the quantification of end-diastolic and end-systolic volumes, stroke volumes, ejection fraction, and left ventricular mass, indexed to body surface area where necessary. Three-chamber SSFP cine images were used to measure linear dimensions of the left ventricle. Four-chamber cine images were used to obtain biatrial areas in end-systole.

Pre-contrast T1 and T2 maps as well as T2W images were initially assessed visually for the presence of visible hyperintense areas. Pre-contrast T1 and T2 relaxation times, T2 signal intensity (T2 SI) ratio, and post-contrast T1 relaxation times were calculated from a 0.7 cm^2^ region of interest (ROI) placed in the mid-ventricular short-axis slice in the mid-section of the interventricular septum, avoiding the right/left ventricle insertion points, or placed in the areas of visible hyperintensity if present. Caution was taken not to include LGE areas in the measurements and not to include blood pool in the ROI. For blood pool pre- and post-contrast T1 calculations, a ROI of the same size was placed at the same level in the ventricular cavity but separate from the papillary muscles or trabeculations. For T2 SI ratio, a ROI in the skeletal muscles of the chest of the same size was used. Extracellular volume was calculated using the previously validated equation [18].The presence and location of LGE was assessed visually. Abnormal native T1 and T2 values were defined as greater than 2 standard deviations above the mean in the control group. An increased myocardial T2 SI ratio was defined as a signal intensity ratio of the LV myocardium to skeletal muscle ≥ 2.0 [19]. Acute myocarditis was defined according to the updated Lake Louise criteria using a combination of T1- and T2-based criteria [20].

### 2.3. ECG Holter Monitoring

Cardiac functional changes were assessed by the 24-h ECG Holter monitoring and were interpreted by an experienced cardiologist blinded to other patient’s data.

### 2.4. Control Group

A control group used to compare CMR findings consisted of 10 sex- and age-matched patients who underwent CMR on the same scanner and did not have any structural cardiovascular disease detected. The indications for CMR in this group included mainly suspicion of myocardial disease based on echocardiography, while CMR, ECG, laboratory tests, and medical history were negative.These patients were derived from the database of the MR Unit at the National Institute of Cardiology in Warsaw. We decided not to include healthy volunteers as a control group because administration of gadolinium contrast for full comparisons to the studied group was deemed unjustified.

### 2.5. Statistical Analysis

All results for categorical variables were presented as a number and percentage. Continuous variables were expressed as median and interquartile range (IQR). Either the chi-square test or the Fisher exact test were used for the comparison of categorical variables when appropriate. Mann–Whitney test for unpaired samples was applied to compare two continuous variables between study group and controls, and Kruskal–Wallis test was used to test differences between more than two groups of continuous variables in the study group. To assess the correlation between continuous variables, the Spearman test was applied. All tests were two-sided with a significance level of *p* < 0.05. Statistical analyses were performed with MedCalc statistical software 10.0.2.0 (Ostend, Belgium).

## 3. Results

Baseline characteristics of the study group of 18 patients with RA are presented in Table 1. Themedian ageof RA patients was 41 years (IQR: 38–45), with a female preponderance (83%). Median time from RA diagnosis was 8.5 years (IQR 5–11), but the median time from symptom onset until diagnosis and starting treatment was relatively short (three months, IQR 2–6). At the time of the study, all patients were on treatment, mainly with methotrexate and leflunomide. Most patients were in remission or had a low disease activity (61%) as demonstrated by their DAS28 score (median 2.9–3.0). Evidence of radiographic damage, typical for RA, was present in 67% patients (median Steinbrocker’s stage III, which includes frequent erosion of joint surfaces). 

There was no difference between the study group and control group in terms of the volume of heart cavities, ejection fraction of the left and right ventricles, left ventricular mass, or left ventricular wall thickness (Table 2). There was also no noticeable pericardial effusion in either group. However, pre-contrast T1 time and T2 time were higher in patients with RA in comparison to controls (Table 2 and Figure 1A,B).While pre-contrast T1 time was above cut-off value in only one patient with RA (6%) and in none of the controls, T2 time was above cut-off value in seven patients with RA (39%) in comparison to none of the controls (*p* = 1.0 and *p* = 0.003, respectively). This was accompanied by a trend towards a higher T2 SI ratio in patients with RA (Table 2 and Figure 1C). T2 SI ratio was above cut-off value in six patients with RA (33%) and in none of the controls (*p* = 0.06), altogether showing signs of myocardial oedema in over a third of RA patients. On the contrary, patients with RA had lower T1 post-contrast values in comparison to controls, which in summary, transferred into similar ECV in both groups and therefore demonstrated a lack of diffuse fibrosis in patients with RA. Examples of myocardial tissue changes observed in patients with RA in comparison to controls are presented in Figure 2.

There were no signs of late gadolinium enhancement (LGE) in either group except for one patient with RA who was found to have lateral wall subendocardial scarring typical of previous myocardial infarction (MI) (Figure 3). The patient was a 41-year-old male with new onset RA (three months of treatment), with no history of cardiac disease and without a family history of early MI. He was overweight, smoked tobacco, and had an impaired fasting glucose. His electrocardiogram did not demonstrate Q waves in the lateral leads. He was started on low-dose aspirin (75 mg) and a high-dose statin (atorvastatin 40 mg) and referred for computed tomography of the coronary arteries (CTCA) and exercise tolerance testing. His exercise tolerance test was normal, but CTCA revealed atherosclerotic plaques with maximal 20–30% stenosis in the coronary arteries. His control LDL-cholesterol after three months of statin treatment was 61 mg/dL.

As none of the patients with RA fulfilled the modified Lake Louise criteria, there were no cases of acute myocarditis. Lack of LGE of mid-wall or subepicardial location ruled out persistent changes due to previous myocarditis. There was no correlation between markers of disease severity and markers of oedema observed on CMR in patients with RA. However, patients with increased T2 (≥50 ms) were more likely to have X-ray lesions (*p* = 0.02) and a longer duration between symptom onset and diagnosis (*p* = 0.02).In addition, we found no significant arrhythmias on 24-h ECG Holter monitoring in RA patients. We observed only single supraventricular and ventricular premature beats within normal limits (up to 50 of ventricular and 100 of supraventricular premature beats) [21,22].

## 4. Discussion

Subclinical inflammation in RA can occur in many organs, including the heart, where traditional ECG and echocardiography may not show any changes. CMR is an imaging technique able to detect inflammatory heart disease. According to the European Society of Cardiology (ESC) and American Heart Association (AHA) statements, CMR is a useful tool in the evaluation of suspected myocarditis [23,24,25]. CMR allows non-invasive visualisation of the pathophysiologic processes seen in myocarditis including myocardial oedema, hyperaemia and capillary leak, and cardiomyocyte necrosis, typically in a non-ischemic pattern on LGE imaging, which fulfil Lake Louise MR diagnostic criteria of myocardial inflammation [19,20,26,27]. CMR provides highly accurate information about cardiac morphology [28]. Moreover, CMR has similar accuracy to endo-myocardial biopsy with the advantage of being a non-invasive method of diagnosis [29].

Our study is one of the first ones to combine parametric imaging (T1 and T2 mapping, ECV calculation), T2 SI ratio calculation, and LGE to assess myocardial abnormalities with CMR in asymptomatic RA patients. The new sequences allow advanced myocardial tissue characteristics of young RA patients. Our data showed that signs of myocardial oedema were present in up to 39% of RA patients, but there was a lack of diffuse fibrosis. There were also no signs of LGE except for one patient with RA who was found to have subendocardial scarring typical of prior silent MI.

A single previous study has used both T1 and T2 myocardial mapping on a group of patients with RA. Greulich et al. demonstrated higher T1, ECV, and T2 values compared to controls [30]. In our study, patients with RA had a higher pre-contrast T1 time but lower T1 post-contrast values in comparison to controls, which in summary transferred into similar ECV in both groups. In addition, Greulich et al. found that 18% of RA patients had a non-ischaemic-type LGE pattern, which is not consistent with our results. The difference might be explained by inclusion of an older population with a longer duration of RA, and presence of cardio-vascular disease (CVD) risk factors, such as hypertension and diabetes, with which more myocardial changes can be expected. As with our study, Greulich et al. emphasised that the most significant differences were observed for T2 values, which did not correlate with disease activity (DAS-28).The absence of correlation could be caused by low to moderate disease activity in both studies. Greulich et al. demonstrated an association between T2 values and disease duration. In our study, patients with increased T2 values (≥50 ms) were more likely to have X-ray erosions and a longer duration between symptom onset and diagnosis. Bone erosions may indicate a more aggressive course of the disease in the past. In the 2016 update of the EULAR recommendations, it was reported that RA patients who are referred to a specialist within three months show better outcomes in terms of drug-free remission and radiographic damage [31]. Lehmonen et al. detected with CMR tagging that early treatment in RA can maintain or improve myocardial function [32]. Our results suggest that early diagnosis and treatment initiation may prevent not only joint destruction but also myocardial damage. Additionally, we and Greulichet al. did not observe differences between the study group and control group in terms of left ventricular size or ejection fraction, as reported also by previous studies [8,12,33,34].

Other studies in asymptomatic RA patients did not use such a comprehensive approach to CMR imaging and were limited to T1 mapping only. For example, Bradham et al. used pre-contrast T1 mapping, which was similar in patients with RA and control subjects [12]. LGE was present in two of 59 (3%) RA patients, and ECV was significantly lower in RA patients compared to controls. Evidence from this study suggested that the absence of fibrosis on CMR in RA patients was due to low to moderate disease activity, which is in line with our findings. Ntusi et al. found focal fibrosis on LGE in 46% of RA patients and none of the controls. Pre-contrastT1 mapping and ECV were higher in RA and correlated with RA disease activity. Compared to our current study, patients were older and had hypertension, diabetes, and slightly higher DAS28-CRP scores. Moreover, Ntusi et al. did not find a significant difference in the overall global myocardial T2 SI ratio between RA patients and controls, but T2 mapping, which has been demonstrated to have better sensitivity, was not used in that study [35].

Kobayashi et al. reported that 32% of RA patients had LGE and 12% had an increased T2 SI ratio [36]. They found that CMR findings indicating myocardial inflammation and fibrosis correlated with RA disease activity. In other studies, which were performed without the use of CMR T1/T2 mapping sequences, focal and diffuse myocardial fibrosis and oedema were frequently seen in RA patients but not in controls [7,9,33,37]. Abnormal CMR findings were usually associated with higher RA disease activity. Patients in these studies were older, had higher DAS28-CRP scores, and in some studies, CVD risk factors were not excluded.

Heart involvement in RA is often clinically silent and can manifest after a long subclinical phase [38]. The risk of MI in RA patients is about 1.6 compared to the general population [39]. RA is associated with the same risk of MI as diabetes mellitus, and the risk of MI in RA is similar to the risk in non-RA subjects 10 years older [40]. We detected subendocardial scarring in a 41-year-old male with newly diagnosed RA (three months into treatment) typical for silent myocardial infarction. He had high disease activity (DAS28 CRP—5,33), bone erosions, and a long duration from symptom onset to diagnosis. In follow up, this patient did not achieve complete remission after one year of treatment (methotrexate, sulfasalazine, hydroxychloroquine, and prednisolone), and he has been qualified for biological treatment. This case further confirms the importance of early treatment initiation.

Myocardial oedema and fibrosis typical for myocarditis may cause conduction defects and arrhythmias [41]. However, we found no significant arrhythmiason 24-h ECG Holter monitoring. We observed only single supraventricular and ventricular premature beats within normal limits. Tłustochowicz et al. found cardiac arrhythmias in 50% patients with RA, but their prevalence was similar to that in the controls [42]. Again, this could be explained by the fact that patients in their study were significantly older and had CVD or CVD risk factors, which we excluded.

The clinical significance of our findings is so far unknown. In previous studies, isolated myocardial oedema without fibrosis has not been shown to independently affect prognosis in patients with suspected myocarditis [43,44]. However, the impact of the long-term presence of myocardial oedema (as may be suspected in patients with RA) on prognosis has not been studied yet.

We are aware that our research may have limitations. For example, the number of RA patients is relatively small. However, many studies had a similar number of patients. Further, we did not have clinical endpoints and longitudinal follow up, and therefore, we were not able to assess the clinical impact of our findings.

## 5. Conclusions

Cardiac magnetic resonance features of myocardial oedema without signs of myocardial fibrosis were found in 39% of young RA patients with no known heart disease or cardiac symptoms. The presence of myocardial oedema was associated with X-ray erosions and a longer duration between symptom onset and diagnosis, which suggests that early diagnosis and effective treatment initiation may prevent not only joint destruction but also myocardial damage. However, the clinical significance of the observed early myocardial changes in young RA patients requires additional studies.

## Figures and Tables

**Figure 1 diagnostics-11-02290-f001:**
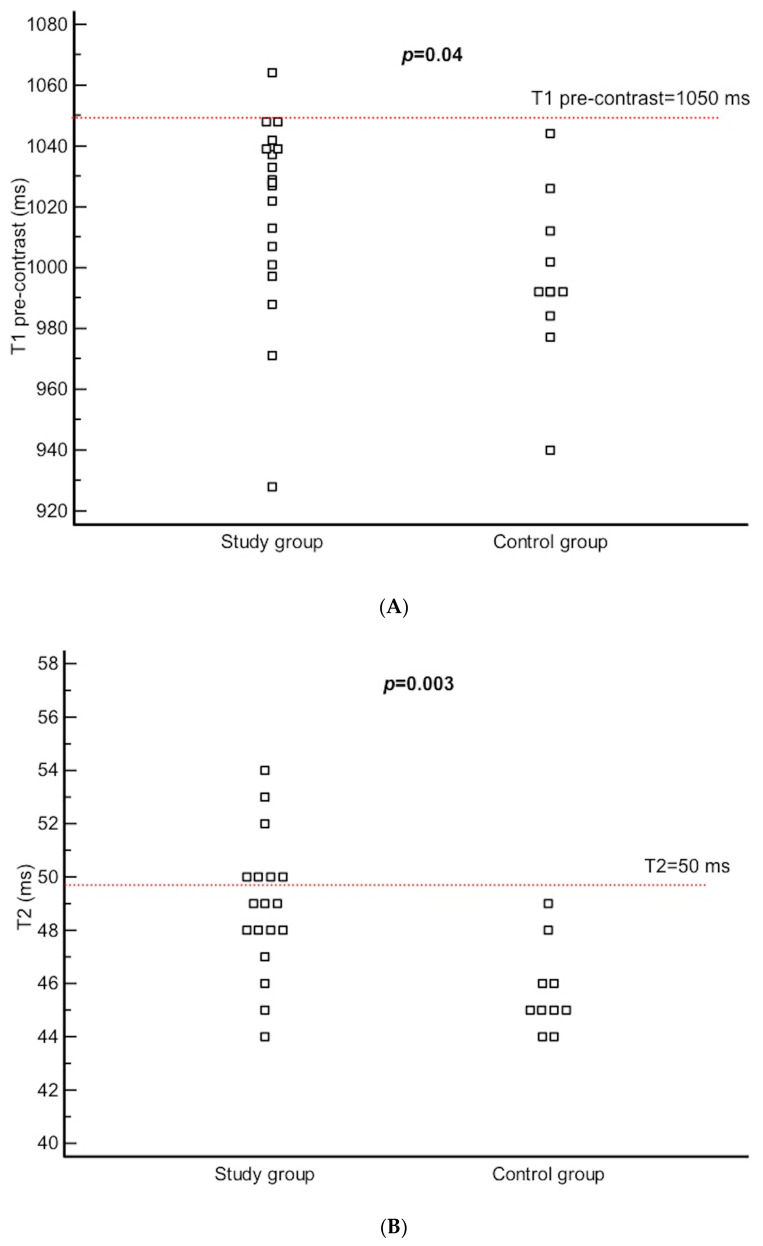
(**A**) Distribution plot comparing T1 pre-contrast times in patients with RA and controls. (**B**) Distribution plot comparing T2 times in patients with RA and controls. (**C**) Distribution plot comparing T2 SI ratio in patients with RA and controls. Cut-off points for normal values are marked with a dotted red lines.

**Figure 2 diagnostics-11-02290-f002:**
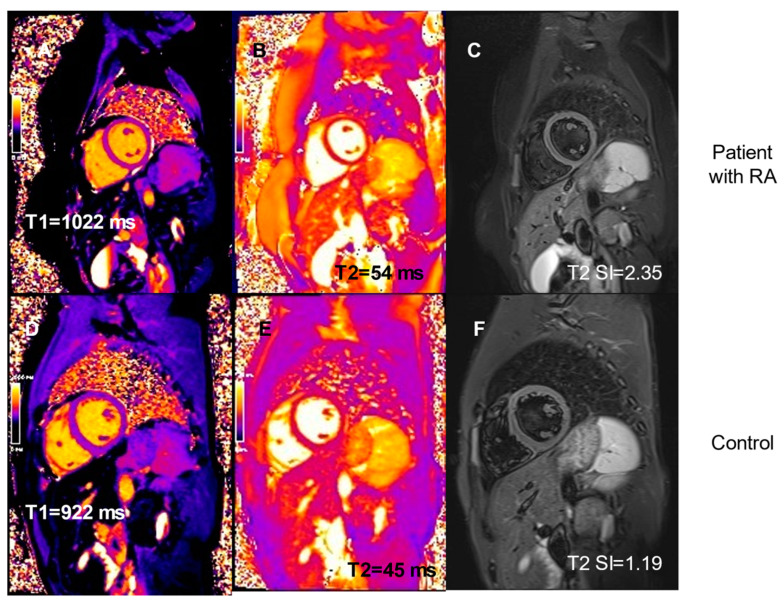
Examples of cardiac magnetic resonance images in mid-ventricular short axis views demonstrating signs of myocardial oedema in patients with RA and lack of those in controls: T1 pre-contrast time (**A**,**D**), T2 time (**B**,**E**), and T2 SI ratio (**C**,**F**).

**Figure 3 diagnostics-11-02290-f003:**
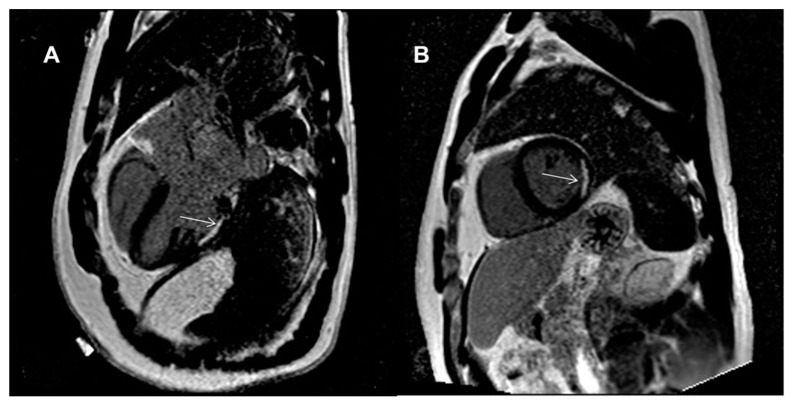
Example of subendocardial LGE in the lateral wall encompassing 50–75% of the myocardial wall thickness and causing myocardial akinesis in that segments with preserved left ventricular ejection fraction found in the 41-year-old male without previously known myocardial infarction. (**A**) 3-chamber view; (**B**) short axis mid-ventricular view (LGE is marked with an arrow).

**Table 1 diagnostics-11-02290-t001:** Baseline characteristics of the studied group.

	Study Group (*n* = 18)
Age, median (IQR), y	41 (38–45)
Female sex, *n* (%)	15 (83)
BMI, median (IQR), kg/m^2^	23.7 (20.3–26.6)
Time from RA diagnosis, median (IQR), y	8.5 (5–11)
Time from symptom onset until diagnosis (treatment start), median (IQR), months	3 (2–6)
ESR, median (IQR), mm/h	10 (5–18)
CRP, median (IQR), mg/dL	4 (4–5)
Positive RF, *n* (%)	15 (83)
Positive CCP, *n* (%)	16 (89)
Swelled joints, median (IQR), number	0.5 (0–3)
Painful joints, median (IQR), number	2 (0–6)
VAS, median (IQR)	32 (11–53)
DAS28 ESR, median (IQR)	3.0 (2.2–4.0)
DAS28 CRP, median (IQR)	2.9 (1.6–4.0)
Disease activity, *n* (%)	
− remission	6 (33)
− low	5 (28)
− moderate	5 (28)
− high	2 (11)
X-ray erosion, *n* (%)	12 (67)
X-ray Steinbrocker scale, median (IQR)	3 (2–3)
HAQ, median (IQR)	0.63 (0.13–1)
Smoking, *n* (%)	
− current	2 (11)
− former	6 (33)
Treatment, *n* (%)	
− steroids	2 (11)
− methotrexate	13 (72)
− leflunomide	5 (28)
− sulfasalazine	2 (11)
− chloroquine	1 (5)
− tocilizumab	2 (11)
− anti-TNF	1 (5)

BMI, body mass index; CCP, citric citrullinated peptide; CRP, C-reactive protein; DAS28, disease activity score; ESR, erythrocyte sedimentation rate; HAQ, Health Assessment Questionnaire; IQR, interquartile range; RA, rheumatoid arthritis;RF, rheumatoid factor; TNF, tumour necrosis factor; VAS, visual analogue scale.

**Table 2 diagnostics-11-02290-t002:** Cardiac MR findings.

	Study Group (*n* = 18)	Control Group (*n* = 10)	*p*-Value
Age, median (IQR), y	41 (38–45)	37 (36–45)	0.17
Female sex, *n* (%)	15 (83)	8 (80%)	1.00
LVEDVI, median (IQR), mL/m^2^	77 (68–85)	84 (80–85)	0.19
LVESVI, median (IQR), mL/m^2^	31 (27–34)	31 (30–33)	0.77
LVSVI, median (IQR), mL/m^2^	49 (45–51)	51 (50–53)	0.19
LVEF, median (IQR), %	61 (59–62)	62 (61–63)	0.36
LVMI, median (IQR), kg/m^2^	60 (51–66)	60 (56–65)	0.63
RVEDVI, median (IQR), mL/m^2^	79 (71–88)	85 (80–86)	0.10
RVESVI, median (IQR), mL/m^2^	32 (27–36)	35 (31–37)	0.26
RVSVI, median (IQR), mL/m^2^	48 (42–53)	51 (48–54)	0.46
RVEF, median (IQR), %	61 (56–66)	59 (57–61)	0.53
LAA, median (IQR), cm^2^	23 (20–24)	24 (23–26)	0.11
RAA, median (IQR), cm^2^	21 (18–23)	23 (22–24)	0.12
IVSd, median (IQR), mm	9 (7.5–9)	9 (8–9)	0.72
PWd, median (IQR), mm	8 (7–8)	8 (8–8.5)	0.29
T1 pre-contrast, median	1028 (1001–1040)	992 (984–1012)	0.04
(IQR), ms			
T1post-contrast, median (IQR), ms	507 (475–550)	586 (565–593)	0.0002
T2, median (IQR), ms	49 (48–50)	45 (45–46)	0.003
T2 SI, median (IQR)	1.92 (1.80–2.13)	1.82 (1.47–1.88)	0.06
LGE, *n* (%)			-
− subendocardial	1 (6)	0 (0)	
− mid-wall	0 (0)	0 (0)	
− subepicardial	0 (0)	0 (0)	
ECV, median (IQR), %	27 (25–30)	27 (25–28)	0.50
Pericardialeffusion, *n* (%)	0 (0)	0 (0)	-

ECV, extracellular volume; IQR, interquartile range; IVSd, interventricular septal diameter; LAA, left atrial area; LGE, late gadolinium enhancement; LVEDVI, left ventricular end-diastolic volume index; LVEF, left ventricular ejection fraction; LVESVI, left-ventricular end-systolic volume index; LVMI, left ventricular mass index; LVSVI, left ventricular stroke volume index; PWd, posterior wall diameter; RAA, right atrial area; RVEDVI, right ventricular end-diastolic volume index; RVEF, right ventricular ejection fraction; RVESVI, right ventricular end-systolic volume index; RVSVI, right ventricular stroke volume index; T2 SI ratio, ratio of signal intensity between myocardium and skeletal muscle on T2W image.

## Data Availability

The data presented in this study are available upon request from the corresponding author.

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
