# Peer review of "Early Myocardial Changes in Patients with Rheumatoid Arthritis without Known Cardiovascular Diseases—A Comprehensive Cardiac Magnetic Resonance Study"

_diagnostics, 2021, doi:10.3390/diagnostics11122290_

Round 1
Reviewer 1 Report
The authors performed cardiac magnetic resonance imaging (CMR) in 18 patients diagnosed with RA and compared them to 10 patients with normal CMR results. They found significant differences between groups, especially T2 time above cut-off value and T2 signal intensity ratio above the cut-off value. In my opinion, the main downside of this very interesting study is that the selection of the control patients is not disclosed. Furthermore, control patients were selected according to a “normal CMR”, which could have led to selection bias. It would be better to select control patients according to baseline characteristics.
Minor comments:
- Please present exact values in the abstract (not just “over a third” or “33-39%”)
- Line 129: Had both specialists 13 years of experience?
- Line 251: how was “premature beats within normal limits” defined?
Author Response
Response to Reviewer 1 Comments First of all, thank you for your comments and suggestions that allowed us to greatly improve the quality of the article. Point 1: The authors performed cardiac magnetic resonance imaging (CMR) in 18 patients diagnosed with RA and compared them to 10 patients with normal CMR results. They found significant differences between groups, especially T2 time above cut-off value and T2 signal intensity ratio above the cut-off value. In my opinion, the main downside of this very interesting study is that the selection of the control patients is not disclosed. Furthermore, control patients were selected according to a “normal CMR”, which could have led to selection bias. It would be better to select control patients according to baseline characteristics. Response 1: Inclusion of age and sex matched controls from MR Unit database is a normal practice in clinical studies. Otherwise we could not justify contrast administration in the control group. All control subjects were cleared from suspicion of structural heart disease. The indications for CMR in this group included mainly suspicion of myocardial disease based on low-quality echocardiography with poor acoustic window, while CMR, ECG, laboratory tests and medical history were negative. Minor comments:- Please present exact values in the abstract (not just “over a third” or “33-39%”)
- We've corrected that.
- Line 129: Had both specialists 13 years of experience?
- Yes both specialists had 13 years of experience. We've modified the text accordingly.
- Line 251: how was “premature beats within normal limits” defined?
- We observed only single supraventricular and ventricular premature beats within normal limits (up to 50 of ventricular and 100 of supraventricular premature beats)
Reviewer 2 Report
This is a very interesting study about the use of c-MRI in RA subjects.
Even if with a small sample size and no follow up, results are juicy and very practical!
I want to congratulate you for your written, since english is fluent and methodology is flawless.
I suggest you the following reference about the growing use of c-MRI in modern scientific literature (https://jcmr-online.biomedcentral.com/articles/10.1186/s12968-020-00688-y)
Author Response
Response to Reviewer 2 Comments This is a very interesting study about the use of c-MRI in RA subjects.Even if with a small sample size and no follow up, results are juicy and very practical! I want to congratulate you for your written, since english is fluent and methodology is flawless.I suggest you the following reference about the growing use of c-MRI in modern scientific literature (https://jcmr-online.biomedcentral.com/articles/10.1186/s12968-020-00688-y) Thank you very much for your comments and ideas for improvements. We have taken into your suggestion and implemented it.
Round 2
Reviewer 1 Report
I congratulate the authors for this paper.